# Continuous Fixed-Bed Column Studies on Congo Red Dye Adsorption-Desorption Using Free and Immobilized *Nelumbo nucifera* Leaf *Adsorbent*

**DOI:** 10.3390/polym14010054

**Published:** 2021-12-24

**Authors:** Vairavel Parimelazhagan, Gautham Jeppu, Nakul Rampal

**Affiliations:** 1Department of Chemical Engineering, Manipal Institute of Technology, Manipal Academy of Higher Education, Manipal 576104, Udupi District, India; gautham.jeppu@gmail.com; 2Department of Chemical Engineering and Biotechnology, University of Cambridge, Philippa Fawcett Drive, Cambridge CB3 0AS, UK; nakulrampal@gmail.com

**Keywords:** Congo red dye, *Nelumbo nucifera* leaf adsorbent, fixed-bed column, immobilization, breakthrough curve, desorption

## Abstract

The adsorption of Congo red (CR), an azo dye, from aqueous solution using free and immobilized agricultural waste biomass of *Nelumbo nucifera* (lotus) has been studied separately in a continuous fixed-bed column operation. The *N. nucifera* leaf powder adsorbent was immobilized in various polymeric matrices and the maximum decolorization efficiency (83.64%) of CR occurred using the polymeric matrix sodium silicate. The maximum efficacy (72.87%) of CR dye desorption was obtained using the solvent methanol. Reusability studies of free and immobilized adsorbents for the decolorization of CR dye were carried out separately in three runs in continuous mode. The % color removal and equilibrium dye uptake of the regenerated free and immobilized adsorbents decreased significantly after the first cycle. The decolorization efficiencies of CR dye adsorption were 53.66% and 43.33%; equilibrium dye uptakes were 1.179 mg g^–1^ and 0.783 mg g^–1^ in the third run of operation with free and immobilized adsorbent, respectively. The column experimental data fit very well to the Thomas and Yoon–Nelson models for the free and immobilized adsorbent with coefficients of correlation R^2^ ≥ 0.976 in various runs. The study concludes that free and immobilized *N. nucifera* can be efficiently used for the removal of CR from synthetic and industrial wastewater in a continuous flow mode. It makes a substantial contribution to the development of new biomass materials for monitoring and remediation of toxic dye-contaminated water resources.

## 1. Introduction

Nowadays, environmental pollution is having an adverse effect on humans and ecosystems. The presence of toxic dye contaminants in aqueous streams, resulting from the discharge of untreated dye containing effluents into water bodies, is one of the essential global environmental problems [1]. Synthetic dyes are widely used in textile, leather, paper, food, pigments, plastics, and cosmetic industries to color their final products, and they consume substantial volumes of water in the process [2]. Rapid industrialization has resulted in the increased disposal of colored effluent into the aquatic environment. Color is the most pervasive contaminant amongst the various pollutants of wastewaters [3]. The presence of dye in effluent even at very low concentration is highly observable and undesirable [4]. The amount of dye lost is dependent upon the types of dyes, the method of application and the depth of shade required [5]. The synthetic dye Congo red (CR) is a polar diazo anionic dye that shows a high affinity for cellulose fibers and is widely used in textile processing industries. It is considered as highly toxic due to its metabolism to benzidine, a human carcinogen [6]. The discharge of CR dye effluents into natural resources has led to many problems in human beings, such as cancer, skin allergy, eye and gastrointestinal irritation [7]. Even a low concentration of CR dye causes various harmful effects, such as difficulties in breathing, diarrhea, nausea, vomiting, abdominal and chest pain, severe headache, etc. [8]. It is used as a laboratory aid for testing free hydrochloric acid in gastric contents, in the diagnosis of amyloidosis, as an indicator of pH, and also as a histological stain for amyloid [9]. The industrial effluents containing CR dye must be treated to bring down its concentration to permissible and bearable levels before discharging into water bodies [10]. Therefore, the removal of CR from industrial effluents is of great importance to present-day researchers. Various conventional treatment methods, such as chemical coagulation, electrochemical oxidation, photo-catalytic degradation, ozonation, sonication, Fenton reagent method, membrane separation, and biological degradation, have been used to treat textile effluents with varying degrees of success in dye removal [11]. However, these technologies have several disadvantages, such as high capital and operating costs, inefficiency in dye removal, complexity of the treatment processes, and the need for the appropriate treatment of residual dye sludge [12]. Amongst the above treatment methods, adsorption is one of the most widely adopted techniques because of its low initial investment, rapid process, efficacy, reliability, versatility, greater flexibility in operation and simplicity of design of equipment, applicability on a large scale, and the easy and safe recovery of the adsorbent as well as adsorbate materials [13]. Moreover, the adsorption-based processes permit the removal of toxic substances from effluent without producing any by-products as compared to conventional treatment methods [14]. Liquid-phase adsorption on powdered activated carbon is a widely used method for the removal of color from wastewater due to its better efficacy and excellent adsorption capacity [15]. However, its widespread use is still limited in large-scale applications, mainly because of the high cost of the adsorbent, besides the difficulty in regeneration and final disposal of the spent activated carbon [16]. Numerous alternative materials from plant and agricultural by-products have been studied by various researchers for adsorbing dyes from aqueous solutions. These adsorbents include wheat bran [17], papaya seeds [18], peanut husks [19], coffee husks [20], guava leaves [21], neem leaves [22], and *malesianus* leaves [23]. These waste materials are cost-effective adsorbents, available in abundant quantities, are highly effective, and the regeneration of these adsorbents may not be necessary, unlike activated carbon [24]. However, the use of agricultural waste remains limited due to the insufficient documentation of the treatment of real-time waste water [13].

Most of these batch adsorption studies focus on the use of adsorbent material in powdered form. The use of powdered adsorbent (plant and agricultural waste material) for the successful removal of toxic dye in large-scale process applications is not practical because of its smaller particle size, low mechanical strength, low density and poor rigidity. The regeneration of the adsorbent after adsorption becomes difficult, and hence results in loss of adsorbent after regeneration. These problems may be rectified by immobilizing the adsorbent in a polymeric matrix, which is used as a supporting material [25]. An immobilized adsorbent has the advantages of a better mechanical strength, offering a higher resistance to various chemical compounds, minimal clogging in continuous-flow systems, easier liquid–solid separation, and ease of regeneration and reuse as compared to a free adsorbent [26]. Batch adsorption studies are not sufficient when designing a treatment system for continuous operation. Column studies are important to obtaining the model parameters required for the design of continuous fixed-bed adsorbers in the large-scale treatment of wastewater [27]. A large volume of effluent can be treated continuously using a fixed quantity of adsorbent in an appropriately designed column, resulting in a better quality of the effluent. The column becomes saturated as the available binding sites are occupied by the target adsorbate molecules [28]. A fixed-bed column has several advantages, such as simplicity of operation, high yield, and easy scale-up [29]. However, column studies on the adsorption of CR dye from wastewater using immobilized *N. nucifera* leaf adsorbent is an area that has not been explored much. To the best of our knowledge, there has been practically no work reported describing the potential of using immobilized *Nelumbo nucifera* leaf powder (NNLP) as an adsorbent in continuous fixed-bed operations for the removal of CR dye from aqueous solutions. Therefore, the present research work focusses on the removal of CR from a synthetic and industrial effluent using free and immobilized *N. nucifera* leaf adsorbents separately in a continuous fixed-bed adsorption operation. *N. nucifera* is commonly grown in subtropical and temperate regions [24]. Its leaf, as an agricultural waste material, contains abundant floristic fiber, protein, lignin, cellulose, hemicellulose, flavonoids, alkaloids and major functional groups (hydroxyl, methyl, alkyne, carbonyl and carboxylate), which are responsible for the increase in adsorption of the various toxic pollutants [30]. The *N. nucifera* leaves are cheap, widely available in India, and can easily be cultivated from seeds or vegetative propagation. The objectives of this present study are to investigate the decolorization and desorption efficiency of free and immobilized NNLP adsorbent separately in various runs. The column experimental data are fitted to various mathematical models to predict the breakthrough curve (BTC) and to evaluate the column capacity and kinetic constants of the models.

## 2. Materials and Methods

### 2.1. N. nucifera Leaf Powder Preparation, Chemical Reagents and Analytical Methods

The mature *N. nucifera* leaves were collected from Bastar district in Chhattisgarh State, India. The leaves were dried under sunlight to remove the moisture and ground to fine powder using a pulverizer. The powdered material was thoroughly washed with distilled water several times to remove all the dirt particles and other impurities. Then, the washed material was dried in a hot-air oven at a temperature of 338 K for 8 h, ground, and screened to obtain particles <100 μm in size [31]. The fine powdered material was stored in an airtight plastic container for further use in adsorption experiments. Analytical grade anionic diazo acid dye Congo red with 99.8% purity was supplied by Sigma-Aldrich, Bengaluru, India. All other chemicals and reagents used throughout this study were of analytical grade and were taken from Merck, Mumbai, India. A stock solution of 1000 mg L^–1^ CR dye was prepared by dissolving 1 g of dye powder in 1000 mL of deionized water. Experimental solutions of the required initial dye concentrations ranging from 15 to 50 mg L^−1^ were made by further diluting the stock solution with pH-adjusted deionized water by adding 0.1 N HCl or 0.1 N NaOH. After dilution, the final pH of the dye solution was measured and was found to be 6. The adsorption experiments are carried out at pH 6 due to the stability of CR dye color [17,27]. The powdered material was weighed using a digital weighing balance (Citizen, Mumbai, India). The pH of the dye solution was measured by a digital pH-meter (Systronics 335, Bengaluru, India). After adsorption, the unknown residual CR dye concentration was determined by measuring the absorbance at 498 nm (λ_max_) using a pre-calibrated double-beam UV/visible spectrophotometer (Shimadzu UV-1800, Tokyo, Japan). The prepared free NNLP adsorbent was characterized by particle size, surface area, and pore volume analyses.

### 2.2. Immobilization of NNLP Adsorbent for CR Dye Adsorption

The various matrices used for the immobilization of NNLP adsorbent are calcium alginate gel, polyvinyl alcohol, polysulfone and sodium silicate [26,32].

#### 2.2.1. Immobilization of the NNLP Adsorbent in Calcium Alginate

A slurry of 2% (*w*/*v*) sodium alginate was prepared in hot distilled water at 333 K for 1 h, resulting in a transparent and viscous solution. After cooling, varying quantities (1–10% *w*/*v*) of NNLP adsorbent were added and stirred for 30 min [26]. For the polymerization and preparation of beads, the alginate–NNLP adsorbent slurry was extruded drop by drop into a cold, sterile 0.05 M calcium chloride solution with the help of a sterile 12 mL syringe with a 2 mm inner diameter [27]. The water-soluble sodium alginate was converted into water-insoluble calcium alginate entrapped on NNLP adsorbent beads via treatment with calcium chloride solution. The beads were hardened by re-suspending them into a fresh cold 0.05 M calcium chloride solution for 24 h with gentle agitation [33].

#### 2.2.2. Immobilization of the NNLP Adsorbent in Polyvinyl Alcohol

A polyvinyl alcohol–sodium alginate slurry in a weight ratio of 2:1 was prepared in hot distilled water at 333 K for 1 h in a beaker. Various quantities (1–10% (*w*/*v*)) of NNLP adsorbent were added to the slurry and the mixtures were stirred for 30 min. Beads were prepared as mentioned above in Section 2.2.1 for polyvinyl alcohol–alginate–NNLP adsorbent slurry in 4% (*w*/*v*) cold calcium chloride solution. The prepared beads were re-suspended in a fresh cold 4% (*w*/*v*) calcium chloride solution for 6 h with gentle agitation to increase the hardness. After 6 h of stabilization, the beads were subjected to three cycles of freezing at <275 K and thawing at 301 K to get spherical beads [26,27].

#### 2.2.3. Immobilization of the NNLP Adsorbent in Polysulfone

A 10% (*w*/*v*) polysulfone solution was prepared in dimethylformamide. Various amounts of the NNLP adsorbent (1–10% (*w*/*v*)) were mixed with the polysulfone slurry. Beads were prepared by following the above procedure as mentioned in Section 2.2.1. The slurry was polymerized in distilled water and the immobilized beads were cured for 16 h in distilled water [26,27].

Various immobilized beads were washed with distilled water and kept in an oven at 323 K for 24 h to remove moisture. Finally, the resultant bead diameter was determined experimentally.

#### 2.2.4. Immobilization of the NNLP Adsorbent in Sodium Silicate

A 6% (*w*/*v*) sodium silicate solution was prepared with distilled water in an Erlenmeyer flask. The sodium silicate solution was added dropwise into 15 mL of 5% (*v*/*v*) sulfuric acid until the pH reached 2. Various quantities of NNLP adsorbent powder (1–10% (*w*/*v*)) were dissolved in 2% (*v*/*v*) acetic acid. Then, 50 mL of this adsorbent dissolved solution was added drop by drop to the silicate solution. Next, the solution was mixed for 15 min. The polymeric gel was made by the addition of sodium silicate solution to reach pH 7. The resultant gel was purified using distilled water to eliminate sulfate ions. Then, the NNLP adsorbent-immobilized sodium silicate was air dried in an oven at 333 K and powdered using a mortar and pestle [32].

### 2.3. Batch Experiments with Synthetic CR Dye Wastewater Using Various Immobilized NNLP Adsorbents

The required amounts of different compositions of various immobilized NNLP adsorbents were added to the CR dye solutions of 250 mg L^−1^ concentration. The dye solutions were stirred at 150 rpm for 24 h at 301 K. The effluent samples were collected and analyzed for residual dye concentration in aqueous solution. A suitable polymeric matrix for the immobilization of NNLP adsorbent was selected based on the maximum adsorption efficiency, effectiveness factor and mechanical stability. The % color removal in dye solution and effectiveness factor of various immobilized NNLP are determined using Equations (1) and (2), respectively [26,34].
(1)$$\%\;\mathrm{CR}\;\mathrm{dye}\;\mathrm{color}\;\mathrm{removal}=\frac{\left(C_{o}-C_{e}\right)\times 100\;}{C_{o}}$$
(2)$$\mathrm{Effectiveness}\;\mathrm{factor}=\frac{\%\;\mathrm{adsorption}\;\mathrm{of}\;\mathrm{CR}\;\mathrm{dye}\;\mathrm{by}\;\mathrm{immobilized}\;\mathrm{NNLP}\;\mathrm{adsorbent}\;}{\%\;\mathrm{adsorption}\;\mathrm{of}\;\mathrm{CR}\;\mathrm{dye}\;\mathrm{by}\;\mathrm{free}\;\mathrm{NNLP}\;\mathrm{adsorbent}}$$
where C_o_ and C_e_ are the initial and equilibrium concentrations of CR in the aqueous solution (mg L^−1^).

### 2.4. Column Experiments for Removal of Color from Synthetic Dye Wastewater Using Free and Immobilized NNLP Adsorbent in Various Runs

Continuous-flow fixed-bed adsorption experiments were performed in a Perspex glass column of 2.1 cm inner diameter and 39 cm height for the removal of CR dye from aqueous solution at 301 K using free and immobilized NNLP adsorbent separately [35]. A schematic diagram of the fixed-bed column is given in Figure 1. A rubber cork of 1.5 cm was provided at the top and bottom of the column to support the inlet and outlet pipes. The column was packed with 1.8 cm of glass wool followed by glass beads, 1.5 mm in diameter, both at the top and bottom [36]. A known amount of free (3.72 g) and sodium silicate gel-immobilized NNLP (3.12 g) adsorbent was packed separately in the column to obtain a bed height of 2.5 cm in the first run. Before the experiment started, the adsorbent packed in the column was wetted with distilled water in the upward flow direction using a peristaltic pump (Enertech, India) to withdraw the trapped air between the particles [37]. The peristaltic pump was used to provide a uniform inlet at the bottom of the column. Steady state flow was maintained by measuring the flow rate at the bottom and top of the column. After setting up the column, the water was replaced with an aqueous CR dye solution of known concentration, 15 mg L^−1^, at pH 6, at the required flow rate of 1 mL min^−1^. The treated dye solution was collected at uniform time intervals from the top of the column with the same flow rate as the feed stream and its concentration was measured using a UV/visible spectrophotometer. The experiments were continued until the concentration of treated dye solutions reached the feed concentration of the adsorbate [38].

### 2.5. Reusability of Free and Immobilized NNLP Adsorbents for CR Dye Adsorption in Column Studies

Desorption studies were performed using various desorbing agents, such as methanol, ethanol, butanol, acetone, and 1 M NaOH in separate batches to explore the possibility of the recovery of adsorbent. After desorption using a suitable desorbing reagent, the free and immobilized adsorbents were collected separately by centrifugation, washed with distilled water, and left to dry at 338 K for 8 h [24]. In order to establish the reusability of the adsorbent, consecutive adsorption–desorption cycles were repeated (three times) using the same adsorbent. The regenerated free and immobilized adsorbents were reused separately for three runs in column experiments. The regenerated adsorbent was added to a dye solution of concentration 15 mg L^−1^ at pH 6. The % color removal of each regenerated NNLP adsorbent was tested and compared to the first run. The column experimental procedure was the same as mentioned before in Section 2.4.

### 2.6. Mathematical Description of Adsorption in a Continuous Fixed-Bed Column Study

The continuous adsorption of dye molecules in a fixed-bed column is an effective process for cyclic adsorption/desorption, as it makes use of the concentration difference, known to be the driving force for adsorption, and allows the more efficient utilization of the adsorbent capacity, resulting in a better-quality effluent. The performance of a fixed-bed column is described through the concept of the breakthrough curve (BTC), which is the plot of time vs. the ratio of effluent adsorbate concentration to inlet adsorbate concentration (C_t_/C_o_). The breakthrough time (t_b_) and bed exhaustion time (t_E_) were used to evaluate the BTCs. Effluent volume (V_eff_) can be calculated as follows [39]:V_eff_ = Q t_E_(3)
where Q is the volumetric flow rate (mL min^−1^). The total quantity of dye adsorbed in the column (m_ad_) is calculated from the area above the BTC multiplied by the flow rate and inlet adsorbate concentration. It is represented as
(4)$$m_{\mathrm{ad}}=\left[\;\overset{t_{E}}{\underset{0}{\int }}\left(1-\frac{C_{t}}{C_{o}}\right)\;\mathrm{dt}\;\right]\;C_{o}\;Q$$

The total amount of dye sent through the column (m_total_) is calculated using the following equation [38]:(5)$$m_{\mathrm{total}}=\frac{C_{o}{\;Q\;t}_{E}}{1000}$$

Total dye removal efficiency with respect to flow volume can be also found from the ratio of the quantity of total adsorbed dye to the total amount of dye sent to the column:(6)$$\mathrm{Total}\;\mathrm{dye}\;\mathrm{removal}\;\left(\%\right)=\frac{m_{\mathrm{ad}}}{m_{\mathrm{total}}}\times 100$$

Equilibrium adsorption capacity, q_e_ (mg g^−1^), is defined as the ratio between the total quantity of solute adsorbed (m_ad_) and the mass of free/immobilized dry adsorbent (W) packed in the column, expressed as [27]
(7)$$\mathrm{qe}\;=\frac{m_{\mathrm{ad}}}{W}$$

### 2.7. Mathematical Models Used for the Breakthrough Curve in Fixed-Bed Column Studies

Various mathematical models, such as the Adams–Bohart, bed depth service time (BDST), Thomas, Yoon–Nelson and Wolborska models, are used in this study to analyze the behavior of the free and immobilized NNLP adsorbent–adsorbate system separately, and to evaluate the kinetic model parameters at various runs [40]. These were then used for the design of the column adsorption process and to scale it up for industrial applications. Linear regression analysis was used to determine the kinetic constants [41].

#### 2.7.1. Adams–Bohart Model

The Adams–Bohart model assumes that the rate of adsorption is proportional to the residual capacity of adsorbent and concentration of the adsorbed species, and it is given by the following linear Equation (8) [42]:(8)$$\ln\left(\frac{C_{t}}{C_{o}}\right)={\;K}_{\mathrm{AB}}C_{o}t\;-\frac{K_{\mathrm{AB}}N_{o}\;Z}{U_{o}}$$
where t is the flow time (min), K_AB_ is the Adams–Bohart kinetic constant (L mg^−1^ min^−1^), N_o_ is the maximum saturation concentration (mg L^−1^), Z is the bed height (cm), and U_o_ is the superficial velocity (cm min^−1^). The characteristic operational parameters K_AB_ and N_o_ can be determined from the slope and intercept of the plot of ln $\left(\frac{C_{t}}{C_{o}}\right)$ vs. t.

#### 2.7.2. Bed Depth Service Time (BDST) Model

The BDST model assumes that the rate of adsorption is controlled by the surface reaction between adsorbate and the unused capacity of the adsorbent. This model ignores the intraparticle diffusion and external film resistance, such that the adsorbate is accumulated onto the adsorbent particle surface directly. The service time, t, of a column in the BDST model is given by the following linear equation [43]:(9)$$t\;=\left(\frac{N_{o}\;Z}{C_{o}{\;U}_{o}}\;\right)-\left(\frac{1}{C_{o\;}K}\right)\;\ln\left(\frac{C_{o}\;}{C_{t}}-1\right)$$
where K is the adsorption rate constant (L mg^−1^ min^−1^). The linear plot of t vs. ln $\left(\frac{C_{o}}{C_{t}}-1\right)$ permits the determination of N_o_ and K from the intercept and slope of the plot.

#### 2.7.3. Thomas Model

The Thomas model assumes that the external and pore diffusions are not the rate-controlling steps. The Langmuir kinetics of adsorption are valid, and this follows the pseudo-second-order reversible reaction kinetics with no axial dispersion. The linear form of the Thomas model can be expressed as follows [44]:(10)$$\ln\left(\frac{C_{o}}{C_{t}}-1\right)=\frac{K_{\mathrm{Th}}q_{\mathrm{oTh}}\;W}{Q}-\frac{K_{\mathrm{Th}}C_{o}V_{\mathrm{eff}}}{Q}$$
where K_Th_ is the Thomas rate constant (L min^−1^ mg^−1^) and q_oTh_ is the maximum solid phase concentration of the solute (mg g^−1^). The plot of ln $\;\left(\frac{C_{o}}{C_{t}}-1\right)$ vs. V_eff_ yields a straight line for which the slope K_Th_ and intercept q_oTh_ can be estimated.

#### 2.7.4. Yoon–Nelson Model

A linear form of the Yoon–Nelson model is expressed as follows [45]:(11)$$\ln\left(\frac{C_{t}}{C_{o}-C_{t}}\right)=k_{\mathrm{YN}}\;t-{\tau k}_{\mathrm{YN}}$$
where k_YN_ is the Yoon–Nelson rate constant (min^−1^) and τ is the time required for 50% solute breakthrough (min). The values of k_YN_ and τ can be determined from the slope and intercept of the linear plot of ln $\;\left(\frac{C_{t}}{C_{o}-C_{t}}\right)$ vs. sampling time, t. The adsorption capacity of the column in this model (q_oYN_) can be determined as [25]:(12)$$q_{\mathrm{oYN}\;=\frac{C_{o}\;Q\;\tau }{1000\;W}}$$

#### 2.7.5. Wolborska Model

The Wolborska model describes the adsorption dynamics at low adsorbate concentration BTCs. It assumes that the axial dispersion is negligible at a high flow rate of solution. The linearized form of the Wolborska model is expressed as [46]:(13)$$\ln\left(\frac{C_{t}}{C_{o}}\right)=\frac{\beta_{a}C_{o}\;t}{N_{o}}-\frac{\beta_{a}\;Z}{U_{o}}$$
where β_a_ is the kinetic coefficient of the external mass transfer (min^−1^). A plot of ln $\left(\frac{C_{t}}{C_{o}}\right)$ vs. t is expected to yield a linear curve in which the model constants β_a_ and N_o_ can be evaluated from the intercept and slope, respectively.

### 2.8. Physicochemical Characteristics of Textile Industrial Dye Effluent

The industrial CR dye effluent was collected from Bright Traders, Erode District, Tamil Nadu State, India. The physicochemical characteristics of real industrial CR dye effluent, such as pH, turbidity, total suspended solids, total dissolved solids, biological oxygen demand (BOD), chemical oxygen demand (COD), total alkalinity, total hardness, dissolved oxygen concentration, electrical conductivity, sulfates and chlorides, were analyzed according to standard operating procedures suitable for wastewater samples [47,48]. The optimized value of various process parameters, obtained from the central composite design (CCD), are used to analyze the physicochemical characteristics of the real textile industrial CR dye effluents in batch studies using free and sodium silicate gel-immobilized NNLP adsorbent separately [24]. The methodology used to analyze the physicochemical parameters of real industrial CR dye effluent are reported in Table 1.

## 3. Results and Discussions

### 3.1. Evaluation of a Suitable Matrix for Immobilization of the NNLP Adsorbent in Batch Studies

The Brunauer–Emmett–Teller (BET) surface area and pore volume of the NNLP adsorbent were found to be 4.72 m^2^ g^−1^ and 7.1 mm^3^ g^–1^, respectively, with the average particle size of 93.80 μm. The average diameter of various immobilized NNLP adsorbent beads was found to be 2.25 mm. To optimize NNLP adsorbent loading in each polymeric matrix, immobilized adsorbents were prepared with varying compositions (1–10% (*w*/*v*)) of NNLP adsorbent in each matrix. The results of batch adsorption experiments obtained using different immobilized matrices with varying adsorbent loadings are reported in Figure 2 and Appendix A. Figure 2 shows that the % dye removal increased with increase in adsorbent loading in various polymeric matrices up to an optimal limit, and beyond this it decreased, which may be attributed to the difference in porosity of the beads/gels when a higher dosage of adsorbent was loaded. The beads/gels may be less porous at higher adsorbent concentration. As the adsorbent dose was increased in the different polymeric matrices, the increase in loading may affect the free transport of CR dye anions to the interior binding sites through the formation of a physical boundary layer. This phenomenon can be explained by the agglomeration of adsorbent particles in various polymeric matrices [26]. It may also be due to the screening effect of the denser adsorbent particles at the outer layer of the immobilized matrix [27]. The optimum NNLP adsorbent loading was found to be 5% (*w*/*v*) for calcium alginate, 4% (*w*/*v*) for polyvinyl alcohol, 6% (*w*/*v*) for polysulfone and 3% (*w*/*v*) for sodium silicate. As the adsorbent dose was increased above the optimal limit in various polymeric matrices, the pore volume and surface area of the bead/gel decreased (shown in Table 2). The decrease in pore volume and surface area of the immobilized adsorbent may affect the % color removal [27,52]. The CR dye adsorption efficiencies on various immobilized NNLP adsorbent beads were compared with the adsorption efficiencies on the free adsorbent. The effectiveness factor was determined to be in the order of free NNLP adsorbent (1) > sodium silicate (0.9343) > polysulfone (0.8775) > calcium alginate (0.8583) > polyvinyl alcohol (0.7762), with an initial dye concentration of 200 mg L^−1^. Among the various immobilized polymeric matrices, sodium silicate was chosen as the superior matrix, because it exhibits the highest adsorption efficiency (83.64%) and has the lowest cost among the different polymeric materials studied. The other three matrices were relatively less effective in preferentially adsorbing the CR dye. The calcium alginate, polyvinyl alcohol and polysulfone immobilized adsorbent matrices adhered together to form clumps during both adsorption and desorption cycles in aqueous solutions [26]. This property was encountered with the effective flow of feed solution when the immobilized adsorbent was packed in fixed-bed column reactors. Therefore, further column experiments were conducted using the optimized value of NNLP adsorbent loading in the sodium silicate matrix.

### 3.2. Inference from Desorption Studies and Reusability of Free and Sodium Silicate Gel Immobilized NNLP Adsorbent in Batch Studies

Desorption experiments were performed for the removal of CR from sodium silicate-immobilized NNLP adsorbent, and the results are shown in Figure 3 and Appendix A. They show that the amount of CR dye desorbed decreased with an increasing number of runs. The % desorption in all the runs was determined to be in the order of methanol > ethanol > 1 M NaOH > acetone > butanol, with various desorbing reagents in separate batches. It was found that a maximum of 52.67% of the dye could be desorbed using the solvent methanol in the third run, compared with other desorbing reagents. It was also found that the immobilized NNLP adsorbent was best regenerated using methanol as a solvent. The decrease in desorption efficiency with an increase in the number of runs may be due to the low volume of the desorbing reagent or the lack of agitation speed, which may prevent the further release of bound dye anions into the solvent [21,34]. Desorption may be explained based on electrostatic repulsion between the negatively charged sites of the adsorbent and anionic dye molecules [53]. Due to the strong hydrophilic attraction between desorbing reagent and dye molecules, the adsorbed dye gets solubilized in the solvent methanol [54]. This is the opposite of the electrostatic interaction effect, indicating that ion exchange is probably the major mode of the adsorption process. The very low desorption of dye suggests that chemisorption might be the major mode of dye removal from aqueous solutions by the adsorbent [22]. The dye under consideration is acidic in nature and exhibits significant attraction towards the solvent methanol [55]. Therefore, for the desorption of CR loaded in free and sodium silicate-immobilized NNLP adsorbents, the organic solvent methanol was determined to be the most efficient compared to other desorbing reagents, and hence further batch desorption experiments were carried out using the solvent methanol.

The regenerated sodium silicate gel-immobilized NNLP adsorbent was added to a dye solution of concentration 200 mg L^−1^ in an Erlenmeyer flask. The regenerated adsorbent was tested in the second and third runs. The results obtained from the reusability studies of the decolorization of CR in various runs are shown in Appendix A and Appendix A. They show that, in comparison to the first run, 66.75% adsorption was maintained after 24 h in the second run and 57.84% in the third run of operation.

### 3.3. Reusability of Free and Sodium Silicate Gel-Immobilized NNLP Adsorbents for CR Dye Adsorption in Column Studies

Desorption experiments were conducted to remove CR from free and immobilized NNLP adsorbent individually. After desorption, fixed-bed column studies of CR dye adsorption were carried out in three runs with free and sodium silicate gel-immobilized NNLP adsorbent separately. The column experimental results obtained by using free and immobilized NNLP adsorbents are reported in Table 3. The BTCs (i.e., $\frac{C_{t}}{C_{o}}$ vs. t) for the adsorption of CR onto free and immobilized NNLP adsorbents in various runs are shown in Figure 4 and Figure 5. The results show that the saturation of binding sites for the removal of CR takes place rapidly in the initial phases of the adsorption process. Initially, the effluent was almost free of solute. As the dye solution continued to flow, the amount of solute adsorbed started decreasing because of the progressive saturation of the adsorbate on the binding sites of the NNLP adsorbent. The effluent concentration started to rise until the bed was exhausted [35]. The shape of the BTC followed a sigmoidal trend. The column reached t_B_ and t_E_ faster, and steep BTCs were observed in the second and third runs of operation. The breakthrough time and bed exhaustion time decreased significantly after the first run, showing that the treated effluent volume, total quantity of dye adsorbed in the column, % adsorption and equilibrium dye uptake of the regenerated free and immobilized NNLP adsorbent decreased during the second and third runs as compared to the first run, due to the insufficient desorption of the bound dye anions from the adsorbent surface. The free and immobilized NNLP adsorbent active sites may be blocked with CR dye molecules, leading to a reduction in the number of available binding sites on the adsorbent surface. The % color removal and equilibrium dye uptake using immobilized NNLP adsorbent in all three runs were lower than when free NNLP adsorbent was used. This may be due to the formation of a physical boundary layer around the immobilized matrix. The matrix thereby impedes the accessibility of dye anions to the binding sites of the NNLP adsorbent [26,33]. The decolorization efficiencies of CR dye adsorption were 53.66% and 43.33%; the equilibrium dye uptakes were 1.180 mg g^−1^ and 0.783 mg g^−1^ in the third run of operation with free and immobilized NNLP adsorbent, respectively. Therefore, a decrease in the % color removal was observed when increasing the number of runs. The prepared NNLP adsorbent can be used for up to three runs to adsorb CR dye in aqueous solution after regeneration using the solvent methanol.

### 3.4. Analysis of BTCs for CR Dye Adsorption in Various Runs and Estimation of Kinetic Model Parameters in Different Models

The column experiments were performed at a constant flow rate of 1 mL min^−1^ and a fixed dye concentration of 15 mg L^−1^ at pH 6 in various runs. The column experimental data were fitted to various mathematical models to predict the BTCs and to evaluate the kinetic constants of the models using free and immobilized NNLP adsorbents separately.

#### 3.4.1. Adams–Bohart Model

The Adams–Bohart model kinetic parameters in various runs are reported in Table 4 and Table 5 for free and immobilized adsorbent, respectively. From Table 4 and Table 5, it can be observed that as the number of runs increased, the values of kinetic constant, K_AB_, and maximum saturation concentration, N_o_, decreased. This may be due to a decrease in the number of binding sites available for CR dye adsorption. In addition, the loss of adsorbent in the second and third runs of operation may result in a shorter mass transfer zone, leading to a decrease in the t_B_ [56]_._ The predicted and experimental BTCs obtained for free and immobilized NNLP adsorbent in various runs are shown in Appendix A. Large differences were found between the predicted and experimental BTCs. The linear regression coefficient, R^2^, values were in the range of 0.72–0.88 in different runs, suggesting that this model does not fit the data points very well.

#### 3.4.2. Bed Depth Service Time Model

The BDST model constants in various runs were evaluated and are reported in Table 4 and Table 5 for free and immobilized adsorbent, respectively. Table 4 and Table 5 show that the values of K and N_o_ decreased with increasing numbers of runs. This may be due to the available binding sites being saturated with CR dye molecules [56]. The rate constant K characterizes the rate of solute transfer from a liquid to a solid phase. The predicted and empirical BTCs obtained for free and immobilized NNLP adsorbent in various runs are shown in Appendix A. They show that there is a small deviation between empirical and predicted BTCs. Additionally, the R^2^ values were in the range of 0.94–0.96, indicating that this model does not quite fit the column experimental data in various runs.

#### 3.4.3. Thomas Model

The Thomas model adsorption data in various runs were evaluated, and their values are reported in Table 4 and Table 5 for free and immobilized adsorbent, respectively. Table 4 and Table 5 show that as the number of runs increased, the values of K_Th_ and q_oTH_ decreased significantly. This may be due to the inadequate desorption of the bound dye anions from the solid adsorbent surface and the loss of adsorbent after the first run [27,57]. The predicted and empirical BTCs obtained for free and immobilized NNLP adsorbent in various runs are shown in Figure 6 and Figure 7. The theoretical and empirical data show a similar correlation and breakthrough trend. Additionally, the calculated values of bed capacity, q_oTH_, are close to the experimental values of q_oTH_*,* with a high R^2^ value (>0.98) in various runs, suggesting that this model is valid for CR adsorption. The Thomas model assumes that the external and pore diffusion steps are not the rate limiting steps, that there is no axial dispersion, and that the Langmuir kinetics of adsorption are valid. However, the adsorption process is typically not controlled by chemical reaction kinetics and is often limited by solid–liquid interphase mass transfer, and the effect of axial dispersion may be important at lower flow rates [29,58].

#### 3.4.4. Yoon–Nelson Model

The Yoon–Nelson model constants are evaluated in various runs for free and immobilized adsorbent and the values are reported in Table 4 and Table 5. This shows that the values of adsorption rate constant, K_YN_ and τ, decreased with increasing numbers of runs. This may be because of the inadequate desorption of the adsorbed dye molecules from the solid particle surface and the lack of binding sites on the adsorbent after the first run [27]. Due to the loss in adsorbent active sites during the second and third runs, the adsorbate molecules have less time to diffuse through the adsorbent, which may result in a reduced adsorption rate constant [59]. The predicted and empirical BTCs obtained at various runs for free and immobilized NNLP adsorbent are shown in Figure 8 and Figure 9. The predicted BTCs, calculated bed capacity (q_oYN_) and τ are close to experimental data in various runs, with the standard deviation of *τ* and q_oYN_ being less than 1.536% under the given set of operating conditions. This is proven by the high values of R^2^, ranging from 0.98 to 0.99, which suggests that this model is valid for CR adsorption. The Yoon–Nelson model assumes that the rate of decrease in the probability of adsorption for each solute molecule is proportional to the probability of adsorbate adsorption and the probability of adsorbate breakthrough on the adsorbent. In addition, it assumes that there is negligible or no axial dispersion [27,60]. The advantage of applying the Yoon–Nelson model is that the mathematical application is very direct, and it provides the information of 50% column breakthrough, which enables us to predict the exhaustion period without the need for a long experimental time [35,61].

#### 3.4.5. Wolborska Model

The Wolborska model is applied to column experimental data for the description of the BTCs. The model parameters are determined in various runs for free and immobilized adsorbent, and the values are given in Table 4 and Table 5. From Table 4 and Table 5, we can see that the values of parameters β_a_ and N_o_ are influenced in various runs. It is inferred that as the number of runs increases, the values of model parameters β_a_ and N_o_ decrease. This may be due to the loss of active sites in the adsorbent particle surface for CR dye adsorption [46]. The predicted and experimental BTCs with respect to various runs for free and immobilized adsorbent are shown in Appendix A. As can be seen in Appendix A, there is a large discrepancy between experimental and predicted BTCs. The R^2^ values range from 0.72 to 0.88, suggesting that this model does not appropriately fit the column experimental data in various runs. Therefore, the Adams–Bohart and Wolborska models’ validity may be limited to the range of the process conditions used.

Out of the five kinetic models tested for CR dye adsorption, the Thomas and Yoon–Nelson models show a good fit to the experimental data, with good regression coefficients and very low standard deviations in various runs for both free and immobilized NNLP adsorbent.

### 3.5. Physicochemical Analysis of Textile Industrial CR Dye Effluent in Batch Studies

The physicochemical characteristics of the real and treated textile industrial CR dye effluent have been analyzed and compared with the Central Pollution Control Board (CPCB) standard limits. All of the data were recorded twice, and the average values are reported in Table 6. The optimal conditions for the maximum adsorption efficiency were obtained at an initial pH of 6, a free NNLP adsorbent dosage of 6 g L^−1^, an adsorbent particle size of 42 µm, an agitation speed of 150 rpm, an immobilized NNLP adsorbent dosage of 6.5 g L^−1^, and an immobilized adsorbent particle size of 94 µm. Among all the pollutants in the wastewater, the most important components were COD, BOD, pH, oil, nitrogen, phosphorus, sulfate, and suspended and dissolved solids [62]. From Table 6, we can see that the characteristics of the raw effluent are higher than the standards prescribed by the CPCB. The effluent had high turbidity and was colored with average organic and inorganic loading. The turbidity range (150 NTU) was not within the allowable limit of wastewater, and the sample was identified as having a greater volume of suspended particles [63]. The pH of the raw industrial dye effluent was 8.56, which indicates that the effluents from dyeing industries under study are alkaline in nature, indicating that different types of chemicals, such as sodium hyphochlorite, sodium carbonate, sodium bicarbonate, sodium hydroxide, surfactants and sodium phosphate, were used during the processing steps [48]. These chemicals have tremendous effects on the receiving water because they contain high concentrations of organic matter, non-biodegradable matter and toxic substances. A higher value of BOD (622 mg L^−1^) depletes the dissolved oxygen concentration and kills aquatic fish, while the total suspended solid increases turbidity and allows less light penetration for photosynthetic activities. Most wastewater contains high pH, which is not favorable to the growth of organisms; the dye covers the surface of the water, which reflects the light and further denies aquatic organisms oxygen for their metabolism [64]. The dark color in the wastewater can increase the turbidity of the water body and affect the process of photosynthesis [64]. The higher value of electrical conductivity (5.34 mS cm^−1^) indicates that there are more chemicals and dissolved substances present in the water. Higher amounts of these impurities will lead to a higher conductivity. The value of COD for the given dye effluent was 1845 mg L^−1^, which is higher than the CPCB standard. The higher value of COD is primarily due to the nature of the chemicals employed in the dyeing unit of the textile processing industry. The ratio of BOD:COD obtained from the result was less than 0.5 (0.34), indicating that the effluent contains a large portion of non-biodegradable matter [65]. However, the characteristics of treated industrial CR dye effluent using free and immobilized NNLP adsorbent separately are closer to standard values. This shows that free and immobilized NNLP adsorbent can be efficiently used to remove CR from real industrial dye effluents.

### 3.6. Analysis of Column Experiments with Textile Industrial CR Dye Effluent Using Free and Sodium Silicate Gel-Immobilized NNLP Adsorbents

The fixed-bed column experiments were conducted separately with industrial CR dye effluents using free and sodium silicate gel-immobilized NNLP adsorbents. The inlet adsorbate concentration was measured and it was found to be 215 mg L^−1^. The bed height (2.5 cm) and flow rate (1 mL min^−1^) were kept constant. The column experimental procedure is the same as that given before in Section 2.4. The column experimental results are reported in Table 7. While using free and immobilized adsorbents, the decolorization efficiency of the industrial dye effluent was more than 60%. The efficiencies of solute adsorption were 76.25% and 62.18%; equilibrium solute uptakes were 8 mg g^−1^ and 5.84 mg g^−1^ with free and immobilized NNLP adsorbent, respectively. While using the immobilized NNLP adsorbent, the breakthrough point time, bed exhaustion time, treated effluent volume, equilibrium dye uptake, mass of solute adsorbed and decolorization efficiency were lower than when free NNLP adsorbent was used. The BTCs for the adsorption of solute from industrial dye effluent using free and sodium silicate gel-immobilized NNLP adsorbents are shown in Figure 10. The column reached t_B_ and t_E_ faster, and a steep BTC was observed with the immobilized adsorbent at high inlet adsorbate concentrations. The intensity of the peaks of dye effluent was measured before and after adsorption. The intensity of peaks declined considerably after treatment with both the free and immobilized NNLP adsorbents (shown in Appendix A).

## 4. Conclusions

The adsorbent prepared from *N. nucifera* leaf is an effective adsorbent for the removal of CR from an aqueous solution in continuous-mode experiments. The NNLP adsorbent loading in various polymeric matrices was optimized. The maximum adsorption efficiency (83.64%) of CR occurred using the polymeric matrix sodium silicate. Desorption studies on loaded free and immobilized adsorbent showed that the maximum % of CR dye can be desorbed using the solvent methanol. They show that the % desorption decreased with the increase in the number of runs for all desorbing reagents. The regenerated free and immobilized NNLP adsorbent used separately in column studies showed that the % color removal and equilibrium dye uptake decreased with increases in the number of runs. The regenerated adsorbent can be used effectively for up to three cycles to adsorb CR dye in aqueous solutions, with considerable reductions in adsorption efficiency. The efficiencies of CR dye adsorption were 53.66% and 43.33%; the equilibrium dye uptakes were 1.180 mg g^−1^ and 0.783 mg g^−1^ in the third run of operation with free and immobilized NNLP adsorbent, respectively. The column reached t_B_ and t_E_ faster, and steep BTCs were observed in the second and third runs of operation. Various models were applied to the column empirical data to estimate the breakthrough curves and evaluate the model parameters in various runs using free and immobilized adsorbent separately. While considering the regression coefficient, R^2^, predicted breakthrough curves and adsorption capacity, it could be said that the Thomas and Yoon–Nelson models best fit the real-time column experimental data in various runs. The physicochemical characteristics of the real textile industrial CR dye effluent obtained for free and immobilized NNLP adsorbents were within the limits specified by the CPCB standards. The adsorption of solute from textile industrial CR dye effluent studies shows that the decolorization efficiencies of solute adsorption were 76.25% and 62.18%; the equilibrium solute uptakes were 8 mg g^−1^ and 5.84 mg g^−1^ with the free and immobilized NNLP adsorbent, respectively. While using the immobilized NNLP adsorbent, the equilibrium dye uptake and % color removal were lower than when free NNLP adsorbent was used. The bed was exhausted in a short period (2.25 h) and a steep BTC was observed for an immobilized adsorbent at a high inlet adsorbate concentration. The experimental results conclude that NNLP is an effective adsorbent for the removal of color from synthetic and industrial CR dye effluent in fixed-bed column studies. The higher adsorption efficiency of industrial CR dye effluent suggests that free and immobilized *N. nucifera* leaf fine powders can be used effectively to decolorize other anionic dyes from industrial effluents.

## Figures and Tables

**Figure 1 polymers-14-00054-f001:**
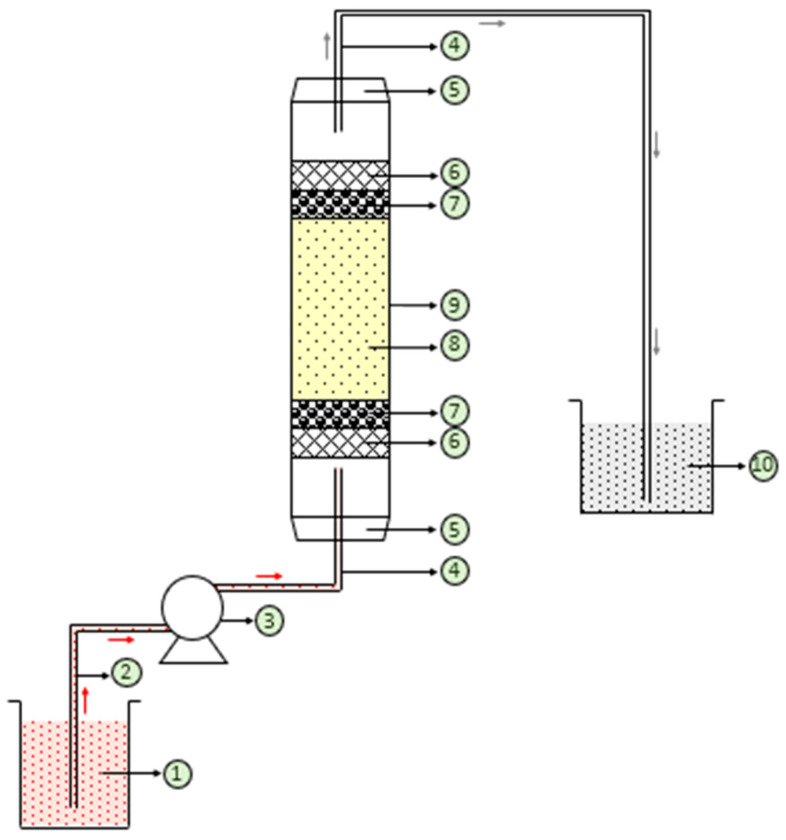
Schematic diagram of the fixed-bed adsorption column. (**1**) Influent CR dye solution; (**2**) silicon tube; (**3**) peristaltic pump; (**4**) glass tube; (**5**) rubber cork; (**6**) glass wool; (**7**) glass beads; (**8**) NNLP adsorbent; (**9**) glass column; (**10**) treated CR dye effluent.

**Figure 2 polymers-14-00054-f002:**
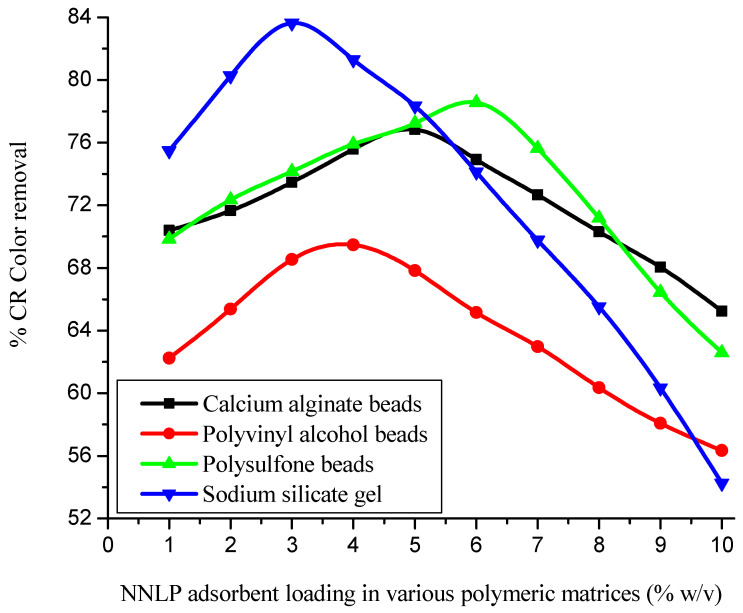
Effect of *Nelumbo nucifera* leaf powder (NNLP) adsorbent loading in various polymeric matrices on CR dye decolorization. (Initial pH: 6; initial dye concentration: 250 mg L^−1^; free NNLP adsorbent particle size: 94 µm; immobilized adsorbent dosage: 6 g L^−1^; agitation speed: 150 rpm; temperature: 301 K; contact time: 24 h)

**Figure 3 polymers-14-00054-f003:**
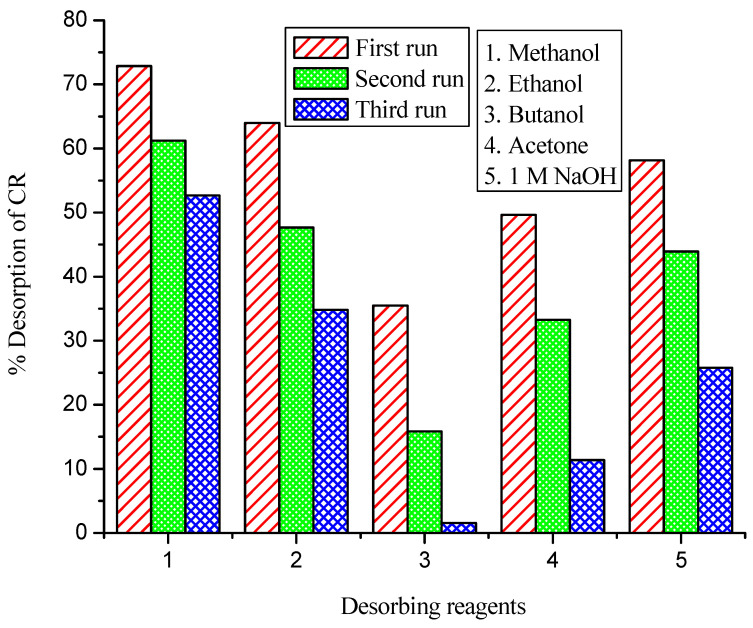
Desorption efficiency of CR dye from sodium silicate gel-immobilized NNLP adsorbent in various runs. (Volume of desorbing reagent: 100 mL; shaking speed: 150 rpm; temperature: 301 K; contact time: 24 h)

**Figure 4 polymers-14-00054-f004:**
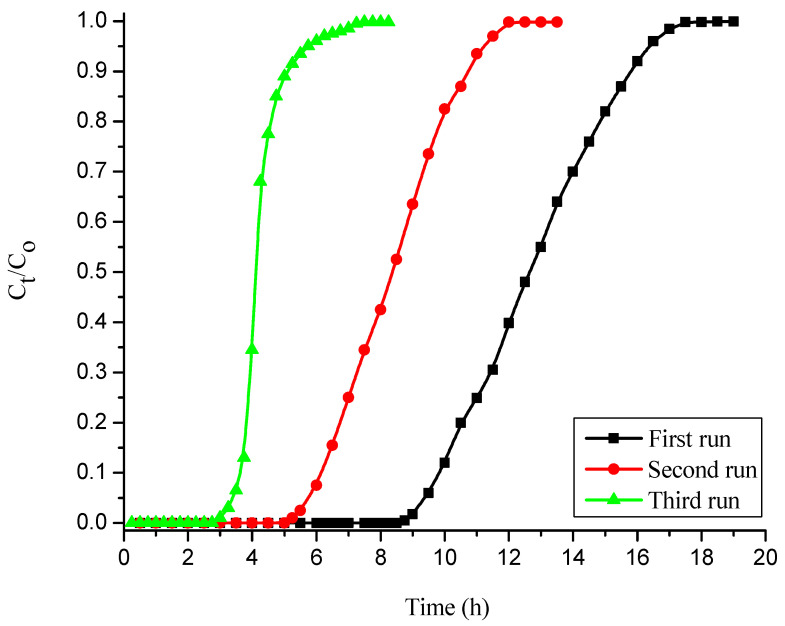
Breakthrough curves (BTCs) for decolorization of CR dye onto free NNLP adsorbent in various runs. (Initial pH: 6; flow rate: 1 mL min^−1^; inlet dye concentration: 15 mg L^−1^; temperature: 301 K)

**Figure 5 polymers-14-00054-f005:**
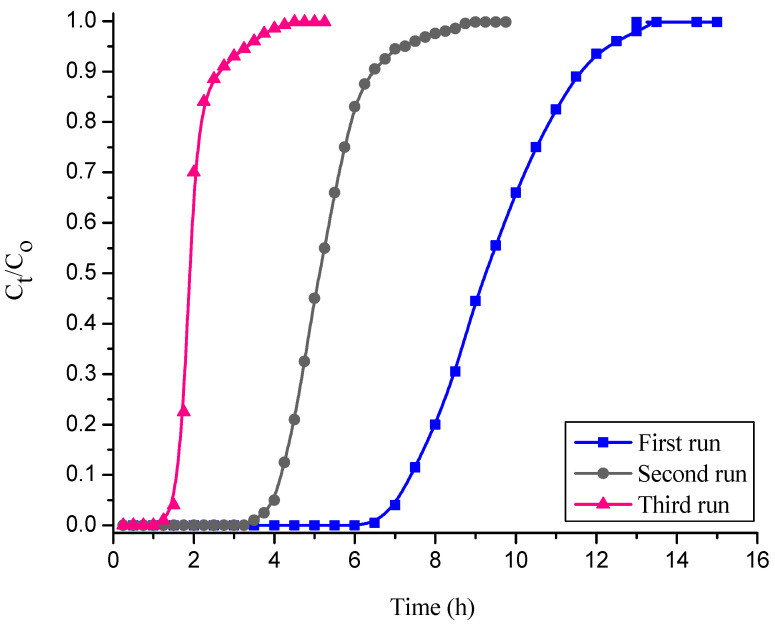
BTCs for decolorization of CR dye by sodium silicate gel-immobilized NNLP adsorbent in various runs. (Initial pH: 6; flow rate: 1 mL min^−1^; inlet dye concentration: 15 mg L^−1^; temperature: 301 K)

**Figure 6 polymers-14-00054-f006:**
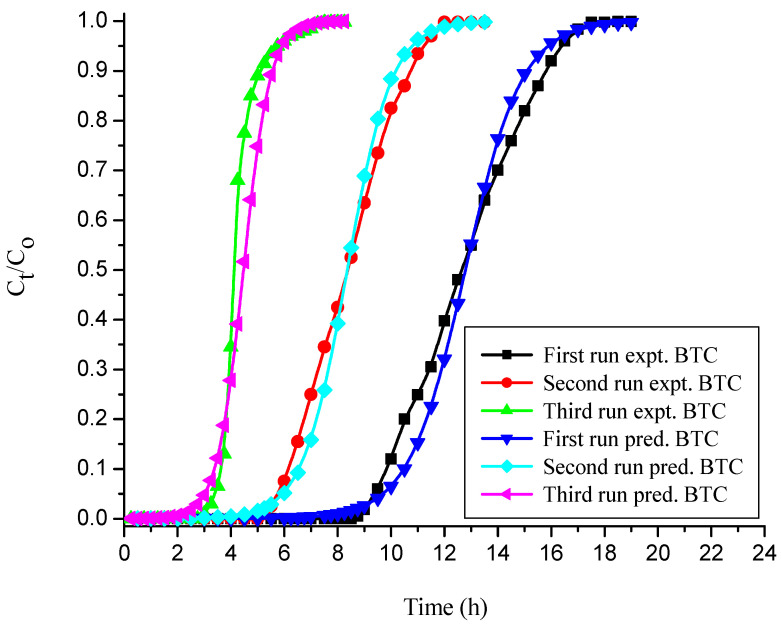
BTCs for experimental vs. simulated Thomas model for the decolorization of CR by free NNLP adsorbent in various runs. (Initial pH: 6; flow rate: 1 mL min^−1^; inlet dye concentration: 15 mg L^−1^; temperature: 301 K)

**Figure 7 polymers-14-00054-f007:**
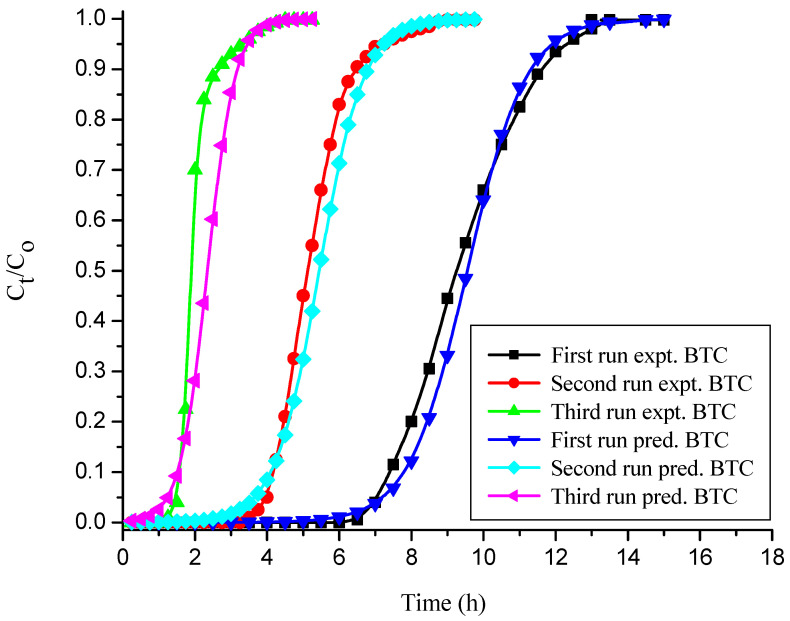
BTCs for experimental vs. simulated Thomas model for the decolorization of CR by sodium silicate gel-immobilized NNLP adsorbent in various runs. (Initial pH: 6; flow rate: 1 mL min^−1^; inlet dye concentration: 15 mg L^−1^; temperature: 301 K)

**Figure 8 polymers-14-00054-f008:**
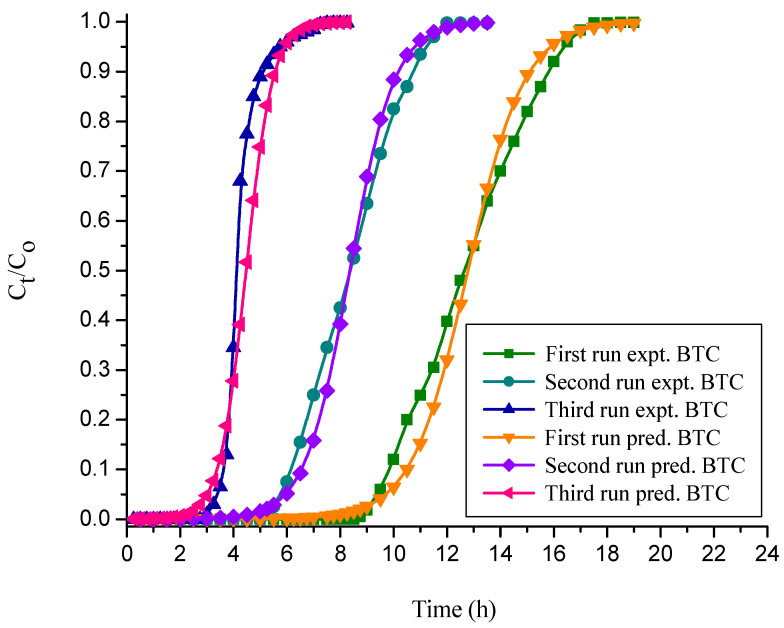
BTCs for experimental vs. simulated Yoon–Nelson model for the decolorization of CR by free NNLP adsorbent in various runs. (Initial pH: 6; flow rate: 1 mL min^−1^; inlet dye concentration: 15 mg L^−1^; temperature: 301 K)

**Figure 9 polymers-14-00054-f009:**
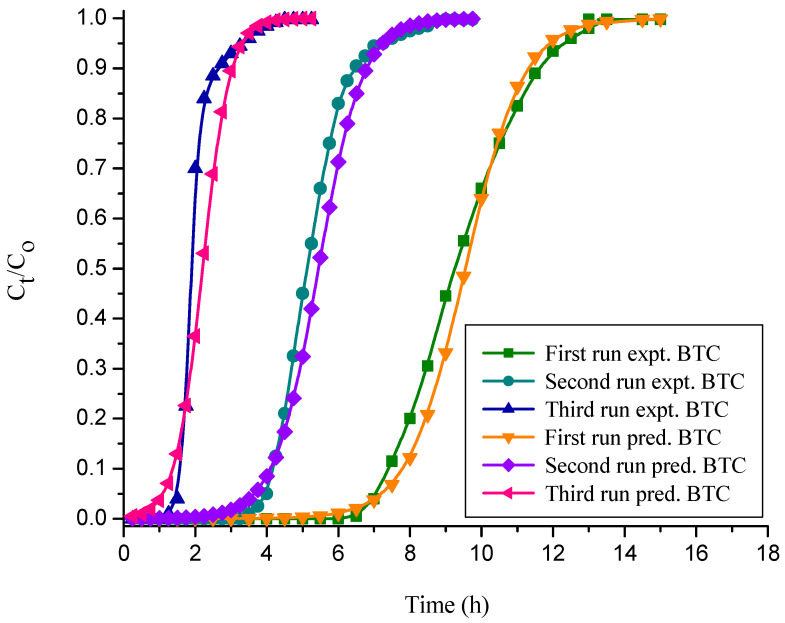
BTCs for experimental vs. simulated Yoon–Nelson model for the decolorization of CR dye by sodium silicate gel-immobilized NNLP adsorbent in various runs. (Initial pH: 6; flow rate: 1 mL min^−1^; inlet dye concentration: 15 mg L^−1^; temperature: 301 K)

**Figure 10 polymers-14-00054-f010:**
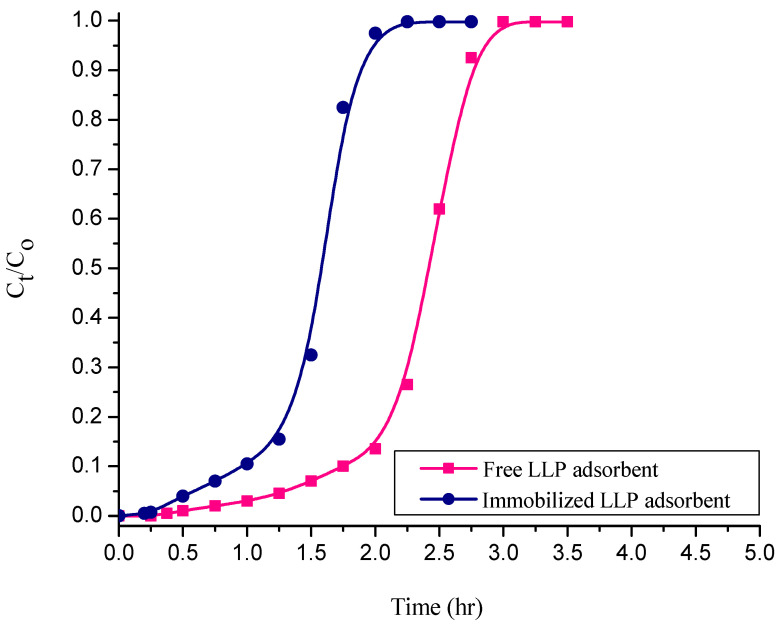
BTCs for adsorption of solute on free and sodium silicate-immobilized NNLP adsorbents with industrial CR dye effluent. (Initial pH: 6; bed height: 2.5 cm; flow rate: 1 mL min^−1^; inlet adsorbate concentration: 215 mg L^−1^; temperature: 301 K)

**Table 1 polymers-14-00054-t001:** Methodology used to analyze the textile industrial Congo red (CR) dye effluent using standard operating procedures [47,48,49,50,51].

Sl. No	Physicochemical Parameters	Method/Instrument
1	pH	Digital pH meter, Systronics
2	Turbidity, NTU	Nephelometric turbidimeter, Systronics
3	Total suspended solids, mg L^−1^	Gravimetric method, oven-drying at 378 K
4	Total dissolved solids, mg L^−1^	Gravimetric method, oven-drying at 378 K
5	Biological oxygen demand, mg L^−1^	Incubating the sample at 303 K for 5 days followed by titration
6	Chemical oxygen demand, mg L^−1^	Closed reflux method
7	Total alkalinity, mg L^−1^	Acid-base titration
8	Total hardness, mg L^−1^	Complexometric titration
9	Dissolved oxygen concentration, mg L^−1^	Dissolved oxygen meter, Systronics
10	Electrical conductivity, mS cm^−1^	Conductivity meter, Digisun
11	Sulfates, mg L^−1^	Titrimetric method
12	Chlorides, mg L^−1^	Argentometric titration

**Table 2 polymers-14-00054-t002:** Physical properties of various polymeric matrices immobilized with NNLP adsorbent.

Various Polymeric Matrices	NNLP Adsorbent Optimum Loading % (*w*/*v*)	Immobilized Adsorbent Characteristics		NNLP Adsorbent Maximum Loading % (*w*/*v*)	Immobilized Adsorbent Characteristics	
		BET Surface Area (m^2^ g^−1^)	Pore Volume (mm^3^ g^−1^)		BET Surface Area (m^2^ g^−1^)	Pore Volume (mm^3^ g^−1^)
Calcium alginate	5	0.65	0.82	10	0.27	0.35
Polyvinyl alcohol	4	0.38	0.53	10	0.13	0.16
Polysulfone	6	0.72	0.94	10	0.34	0.48
Sodium silicate	3	1.84	2.18	10	1.16	1.65

**Table 3 polymers-14-00054-t003:** Column experimental parameters obtained at various runs for the decolorization of CR dye onto free and immobilized NNLP adsorbents.

Adsorbent	Column Parameters	First Run	Second Run	Third Run
Free NNLP adsorbent	Z (cm)	2.50	2.25	2.10
	W (g)	3.72	3.35	3.07
	t_b_ (h)	8.75	5.25	3.00
	t_E_ (h)	17.50	12.00	7.50
	m_ad_ (mg)	12.04	7.053	3.622
	m_total_ (mg)	15.75	10.80	6.75
	V_eff_ (L)	1.05	0.72	0.45
	q_e_ (mg g^−1^)	3.236	2.106	1.180
	% color removal	76.43	65.31	53.66
Sodium silicate gel-immobilized NNLP adsorbent	Z (cm)	2.50	2.10	1.75
	W (g)	3.12	2.69	2.24
	t_b_ (h)	6.50	3.50	1.25
	t_E_ (h)	13.5	9.00	4.50
	m_ad_ (mg)	8.178	4.596	1.755
	m_total_ (mg)	12.15	8.10	4.05
	V_eff_ (L)	0.81	0.54	0.27
	q_e_ (mg g^−1^)	2.621	1.708	0.783
	% color removal	67.30	56.74	43.33

**Table 4 polymers-14-00054-t004:** Column characteristic parameters obtained at various runs for the decolorization of CR onto free NNLP adsorbent.

Free NNLP Adsorbent		Model Parameters		
		First Run	Second Run	Third Run
Adams–Bohart model	K_AB_ (L mg^−1^min^−1^)	4.98 × 10^−4^	3.09 × 10^−4^	1.34 × 10^−4^
	N_o_ (mg L^−1^)	1624.1	1226.7	779.92
	R^2^	0.85	0.88	0.78
BDST model	K (L mg^−1^min^−1^)	1.142 × 10^−3^	8.943 × 10^−4^	6.152 × 10^−4^
	N_o_ (mg L^−1^)	1329.8	966.03	560.18
	R^2^	0.96	0.95	0.96
Thomas model	K_TH_ (mL mg^−1^ min^−1^)	1.069 × 10^−3^	8.264 × 10^−4^	5.839 × 10^−4^
	q_oTH_ (mg g^−1^)	3.236	2.244	1.309
	R^2^	0.98	0.98	0.99
Yoon–Nelson model	K_YN_ (min^−1^)	0.0164	0.0085	0.0057
	τ (min)	766.9	501.3	268.0
	q_oYN_ (mg g^−1^)	3.092	2.244	1.309
	R^2^	0.98	0.98	0.98
Wolborska model	β_a_ (min^−1^)	0.809	0.773	0.658
	N_o_ (mg L^−1^)	1624.1	1226.7	779.92
	R^2^	0.85	0.88	0.78

**Table 5 polymers-14-00054-t005:** Column characteristic parameters obtained at various runs for the decolorization of CR by sodium silicate gel-immobilized NNLP adsorbent.

Immobilized NNLP Adsorbent		Model Parameters		
		First Run	Second Run	Third Run
Adams–Bohart model	K_AB_ (L mg^−1^min^−1^)	4.17 × 10^−4^	2.65 × 10^−4^	1.09 × 10^−4^
	N_o_ (mg L^−1^)	1239.2	938.88	538.04
	R^2^	0.83	0.80	0.72
BDST model	K (L mg^−1^min^−1^)	1.086 × 10^−3^	6.754 × 10^−4^	4.563 × 10^−4^
	N_o_ (mg L^−1^)	994.68	678.16	339.04
	R^2^	0.95	0.95	0.94
Thomas model	K_TH_ (mL mg^−1^ min^−1^)	9.542 × 10^−4^	5.483 × 10^−4^	2.256 × 10^−4^
	q_oTH_ (mg g^−1^)	2.754	1.822	0.944
	R^2^	0.98	0.99	0.98
Yoon–Nelson model	K_YN_ (min^−1^)	0.0118	0.0062	0.0024
	τ (min)	572.9	326.8	132.3
	q_oYN_ (mg g^−1^)	2.754	1.822	0.886
	R^2^	0.98	0.98	0.99
Wolborska model	β_a_ (min^−1^)	0.765	0.642	0.573
	N_o_ (mg L^−1^)	1239.2	938.88	538.04
	R^2^	0.83	0.80	0.72

**Table 6 polymers-14-00054-t006:** Characteristics of real and treated textile industrial CR dye effluent with CPCB acceptable limits.

Sl. No	Physicochemical Parameters	Raw Effluent Value	Treated Effluent Value		CPCB Standard
			Free NNLP Adsorbent	Sodium Silicate-Immobilized NNLP Adsorbent	
1	pH	8.56	8.50	8.52	6–9
2	Turbidity, NTU	150	18	24	<10
3	Total suspended solids, mg L^−1^	134	26	45	<100
4	Total dissolved solids, mg L^−1^	3824	965	1012	<2000
5	Biological oxygen demand, mg L^−1^	622	42	58	<30
6	Chemical oxygen demand, mg L^−1^	1845	238	246	<250
7	Total alkalinity, mg L^−1^	418	87	102	<200
8	Total hardness, mg L^−1^	756	415	420	<300
9	Dissolved oxygen concentration, mg L^−1^	1.12	1.06	1.08	<4.00
10	Electrical conductivity, mS cm^−1^	5.34	4.92	5.13	<2.25
11	Sulfates, mg L^−1^	560	138	165	<250
12	Chlorides, mg L^−1^	1524	236	274	<500

**Table 7 polymers-14-00054-t007:** Column experimental parameters for the adsorption of solute onto free and sodium silicate gel-immobilized NNLP adsorbents with industrial CR dye effluent.

Adsorbent	W (g)	t_b_ (h)	t_E_ (h)	m_ad_ (mg)	m_total_ (mg)	V_eff_ (L)	q_e_ (mg g^−1^)	% Color Removal
Free NNLP adsorbent	3.72	0.375	3.00	29.783	39.06	0.180	8.00	76.25
Immobilized NNLP adsorbent	3.12	0.20	2.25	18.215	29.295	0.135	5.84	62.18

## Data Availability

The data used to support the findings of this study are available from the corresponding author upon request.

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
