# Peer review of "Continuous Fixed-Bed Column Studies on Congo Red Dye Adsorption-Desorption Using Free and Immobilized Nelumbo nucifera Leaf Adsorbent"

_polymers, 2021, doi:10.3390/polym14010054_

Round 1

Reviewer 1 Report

Manuscript entitled “Continuous Fixed-Bed Column Studies on Congo Red Dye Adsorption-Desorption Using Free and Immobilized Nelumbo nucifera Leaf Adsorbent” submitted by Vairavel Parimelazhagan, Gautham Jeppu and Nakul Rampal, can be considered for publication in Polymers Journal, after a major revision.

Here is a list of my specific comments:

  1. General comment 1: The utility of this study should be clearly highlighted in the manuscript.
  2. General comment 2: Pay attention on the originality of this study and avoid, as far as possible, expressions as “these data were discussed in a previous study” (see reference 18 and 30).
  3. Page 2, line 57: “Moreover, the ability of adsorption…”. Add here as reference the paper: The utilization of leaf- based adsorbents for dyes removal: A review, Journal of Molecular Liquids, 276, (2019), 728-747, because it is relevant for this observation.
  4. Page 3, Materials and Methods: Include subsections 2.1-2.5 into one “Chemical reagents and analytical methods” (or similar).
  5. Page 6, line 216: “Desorption characteristics or regeneration…”. This paragraph should be deleted because it is too general.
  6. Page 8, Table 1: Add a reference in the table caption for analytical methods.
  7. Page 10, line 383: “A detailed analysis of the reusability… where [18].” If these results have been already reported, what is the originality of this study???
  8. Page 12, 3.5. Analysis of Column Experiments with Synthetic Dye Wastewater Using Free NNLP Adsorbent: The same observation as above.
  9. Page 12, 3.6. Analysis of BTCs for CR Dye Adsorption in Various Runs with Kinetic Constants in Different Models: This section should be systematized. The experimental results included in this section should be clearly presented and discussed in detail.
  10. Page 17, 3.7. Physico-chemical analysis of textile industrial CR dye effluent in batch studies: In this section: (a) The optimal conditions for the adsorption process should be mentioned. (b) The results included in Table 6 should be more detailed discussed.

Author Response

Dear Sir/Madam,                                                                                                                   

Greetings!

Ref: Manuscript Number: polymers-1485236

Subject: Submission of revised manuscript titled, "Continuous Fixed-Bed Column Studies on Congo Red Dye Adsorption-Desorption Using Free and Immobilized Nelumbo nucifera Leaf
Adsorbent" by P. Vairavel, Gautham Jeppu and Nakul Rampal for possible publication in Polymers

The changes made in the revised manuscript have been done using track changes mode in MS Word. All the authors have verified all the changes. The point-wise responses to reviewer’s comments have been addressed carefully and are attached below, for your kind perusal.

The replies to the reviewer’s comments are given in red colored text. 

Kindly consider the revised manuscript for the possible publication in Polymers.

Section-Polymer Applications

Special issue-Novel Wastewater Treatment Applications Using Polymeric Materials

Warm regards,

1. P. Vairavel (First and Corresponding author)

2. Gautham Jeppu (Co-author 1)

3. Nakul Rampal (Co-author 2)

Reviewer 2 Report

The manuscript is well written, and scientifically sound. Only one minor observation:

The figure caption of Figures 4 and 5 are identical, probably due to an oversight:

line 447: Figure 4. Breakthrough curves (BTCs) for decolorization of CR dye onto free NNLP adsorbent in various runs. (Initial pH: 6; flow rate: 1 mL min-1 ; inlet dye concentration: 15 mg L-1 ; temperature: 301 K.

line 451: Figure 5. Breakthrough curves (BTCs) for decolorization of CR dye onto free NNLP adsorbent in various runs. (Initial pH: 6; flow rate: 1 mL min-1 ; inlet dye concentration: 15 mg L-1 ; temperature: 301 K).

In figure caption section, the figure 4 and 5 have the correct text

Author Response

(The authors gave the same response as above.)

Round 2

Reviewer 1 Report

All my previous remarks and comments have been considered into new version of the manuscript. It means that revised manuscript meets the criteria and in my opinion can be published as original paper in Polymers Journal.